# Multilevel determinants of illness since birth among infants aged 5–9 weeks in Ethiopia: Evidence from the 6-week follow-up of the PMA-ethiopia longitudinal survey (2021–2023)

Amare Mebrat Delie[1]*, Mickiale Hailu[2], Molla Getie Mehari[3], Gizachew Kassahun Bizuneh[4], Berihun Agegn Mengistie[5], Tesfaye Shumet Mekonnen[6]

1 Department of Public Health, College of Medicine and Health Sciences, Injibara University, Injibara, Ethiopia, 2 Department of Midwifery, College of Medicine and health science, Dire Dawa University, Dire Dawa, Ethiopia, 3 Injibara University, College of Medicine and Health Sciences, Department of Medical Laboratory Science, Injibara, Ethiopia, 4 Department of Pharmacognosy, School of Pharmacy, College of Medicine and Health Sciences, University of Gondar, Gondar, Ethiopia, 5 Department of General Midwifery, School of Midwifery, College of Medicine and Health Sciences, University of Gondar, Gondar, Ethiopia, 6 Department of Public Health, School of Public Health, College of Medicine and Health Sciences, Debre-Markos University, Debre-Markos, Ethiopia

* amaremebrat2@gmail.com

## Abstract

### Background

Infant illness in the early weeks of life remains a major public health concern in Ethiopia, contributing to neonatal and infant morbidity and mortality. Identifying individual- and community-level determinants is critical to guide targeted interventions.

### Objective

To assess multilevel determinants of illness since birth among infants aged 5–9 weeks in Ethiopia using data from the PMA-Ethiopia Longitudinal Survey (2021–2023).

### Methods

We analyzed data from 1,960 infants and mothers enrolled in Cohort 2 of the PMA Ethiopia longitudinal survey, conducted across Addis Ababa, Amhara, Oromia, and SNNP regions. A multistage stratified cluster design with sampling weights ensured national representativeness. The outcome was any maternal-reported infant illness since birth. Multilevel binary logistic regression accounted for clustering within mothers and enumeration areas. Adjusted odds ratios with 95% confidence intervals were reported, and model fit was compared using AIC and BIC.

which permits unrestricted use, distribution, and reproduction in any medium, provided the original author and source are credited.

**Data availability statement:** The data underlying this study are third-party data from the Performance Monitoring for Action (PMA)-Ethiopia Longitudinal Survey (2021–2023). The datasets are openly available and were accessed from the PMA website (https://www.pmadata.org/data/available-datasets), which provides free, public access to PMA datasets through the Johns Hopkins University (JHU) Data Repository. No registration or special access privileges were required. All interested researchers can access the same data to replicate the findings of this study.

**Funding:** The author(s) received no specific funding for this work.

**Competing interests:** The authors have declared that no competing interests exist.

**Abbreviations:** AIC: Akaike's Information Criterion; ANC: Ante Natal Care Visit; AOR: Adjusted Odds Ratio; BIC: Bayesian Information Criterion; CI: Confidence Interval; ICC: Intraclass Correlation Coefficient; IPV: Intimate Partner Violence; MOR: Median Odds Ratio; PCV: Proportional Change in Variance; PMA: Performance Monitoring Action.

## Results

Overall, 34.42% (95% CI: 30.23–38.87) of infants experienced illness since birth, while 65.58% (95% CI: 61.13–69.77) remained illness-free. Female infants had lower odds of illness than male infants (AOR = 0.53; 95% CI: 0.35–0.82). Infants born to mothers who experienced pregnancy complications had higher odds of illness (AOR = 1.89; 95% CI: 1.11–3.20), as did infants of mothers with postpartum complications (AOR = 2.33; 95% CI: 1.27–4.27). Infants from households owning insecticide-treated bed nets also had higher odds of illness (AOR = 1.81; 95% CI: 1.10–2.96).

## Conclusion

About one-third of Ethiopian infants experienced illness within 5–9 weeks of life. Female infants were less vulnerable, whereas maternal complications during pregnancy and postpartum markedly increased illness risk. The association with bed net ownership warrants further investigation. Strengthening maternal health services and targeted interventions may reduce early infant morbidity.

## Background

Infant and child mortality are widely recognized as critical indicators of a nation's socioeconomic development and overall quality of life [1]. In response to the global burden of preventable child deaths, Sustainable Development Goal (SDG) 3.2 targets a reduction in neonatal mortality to at least 12 per 1,000 live births and under-five mortality to no more than 25 per 1,000 live births by 2030 [2]. Achieving these targets requires ongoing efforts to systematically document, analyze, and monitor maternal, newborn, and child health outcomes, ensuring that progress is tracked and interventions are effectively guided.

Globally, infectious diseases such as diarrhea, malaria, and pneumonia remain leading causes of morbidity and mortality among children under five, particularly in low- and middle-income countries [3]. Acute respiratory infections alone account for approximately 1.3 million child deaths annually, representing 20% of under-five mortality, while diarrhea contributes to 370,000 deaths, or roughly 9% of the total [4–6]. Pneumonia accounts about 14% of lives in this age group, with an incidence of 1,400 cases per 100,000 children per year [6]. Fever is a common symptom across these illnesses and is often the primary reason for healthcare visits among children in low-resource settings, including Ethiopia [7–9]. It commonly indicates underlying infections such as malaria, acute respiratory infections (ARIs), or gastrointestinal diseases [10]. In Ethiopia, children typically experience between two and nine episodes of fever annually [11].

Childhood illness continues to be a significant public health challenge in Ethiopia. As reported in the 2016 Ethiopian Demographic and Health Survey (EDHS), 7.0% of children under five exhibited symptoms of acute respiratory infections (ARIs), 13.0% had diarrhea, and 14.0% experienced fever in the two weeks preceding the survey [1]. Alarmingly, the 2021 EDHS reported an increase in ARI symptoms to 21.7%, with

diarrhea and fever affecting 12.1% and 14.8% of children, respectively [12]. These findings underscore a persistent and, in some cases, worsening burden of early childhood illnesses [13]. Another previous study revealed a range of early neonatal morbidities, with jaundice requiring phototherapy (32%), respiratory distress (19.1%), and cyanotic episodes (15.7%) among the most frequently reported, followed by conditions such as hypoglycemia, temperature instability, and feeding intolerance [14]. The likelihood of developing complications was found to be over 13 times higher in specific subgroups, underscoring the significance of both modifiable factors such as antenatal steroid use, administration of magnesium sulfate, and mode of delivery and non-modifiable factors like maternal age, parity, twin gestation, and infant gender [15].

While prior studies and national surveys have documented the burden of common childhood illnesses such as fever, diarrhea, and acute respiratory infections in Ethiopia, these studies are predominantly cross-sectional and rely on retrospective caregiver reports [14,16–18]. As a result, they are limited in their ability to capture the early postnatal period, assess temporal patterns of illness, and identify multilevel determinants that operate at both individual and community levels particularly in rural settings. Moreover, existing evidence in Ethiopia largely focuses on under-five children as a broad age group, with limited attention to early infancy, a critical developmental window characterized by heightened vulnerability to illness and mortality. Robust longitudinal data examining illness prevalence and risk factors among infants in the first weeks of life are especially scarce [16,17,19,20].

To address this gap, the present study utilizes nationally representative longitudinal data from the PMA-Ethiopia project (2021–2023) to estimate the prevalence of any illness among infants aged 5–9 weeks and to identify individual- and community-level risk factors. By leveraging a multilevel analytic approach, this study aims to provide critical insights that can inform targeted interventions, optimize resource allocation, and ultimately contribute to the national and global efforts to improve early childhood health outcomes.

## Methods

### Data source

This study is based on secondary data from the six-week follow-up survey of Cohort 2 of the Performance Monitoring for Action (PMA) Ethiopia longitudinal study, conducted between 2021 and 2023. PMA Ethiopia is a five-year collaborative initiative involving Addis Ababa University, Johns Hopkins University, and the Ethiopian Ministry of Health, aimed at generating high-quality data to inform reproductive, maternal, and newborn health (RMNH) policies and programs.

The longitudinal study tracks women from pregnancy through one year postpartum, with data collected at multiple intervals. It is carried out in four major agrarian regions in Amhara, Oromia, Tigray, and the Southern Nations, Nationalities, and Peoples' Region (SNNPR) along with the urban hub of Addis Ababa. While the first cohort (2019–2021) included the Afar region, Cohort 2, which is the focus of this study, did not [21].

### Sampling technique

The study employed a two-stage stratified cluster sampling design, with 162 enumeration areas selected from the Central Statistical Agency's master sampling frame using probability proportional to size, stratified by region and urban-rural residence. Within each enumeration areas, a household census identified women aged 15–49 who were either pregnant or up to 9 weeks postpartum. Eligible women were recruited and participated in baseline interviews. A total of 2,297 women provided informed consent to participate in the six-week follow-up survey. PMA Ethiopia categorizes households in two ways: those identified through household screening as containing a currently pregnant or postpartum woman (0–9 weeks) and those selected for cross-sectional interviews where a panel-eligible woman was later identified. Based on reproductive status at baseline, women were grouped into four categories: currently pregnant, 0–4 weeks postpartum, 5–9 weeks postpartum, and ineligible or non-consenting women [21]. Importantly, 273 women who were already 5–9 weeks postpartum at the time of the baseline interview completed both baseline and six-week follow-up interviews concurrently. These cases are flagged using the "cohort type" variable and represent a unique subgroup for whom complete information

on delivery, neonatal care, birth outcomes, and postnatal care is available. Although household rosters captured data on all members, only panel-eligible women completed the full female questionnaire. Longitudinal variables were formatted in a wide structure, with repeated events assigned numerical indices to ensure efficient tracking and integration across successive survey rounds [21]. The study population was all early infants aged 5–9 weeks in Ethiopia, assessed through interviews with their mothers participating in the PMA Ethiopia Longitudinal Panel Survey.

### Data collection procedures and data quality assurance

Baseline data were collected between October 2021 and January 2022 using standardized, mobile-assisted personal interviews conducted on smartphones or tablets via the Open Data Kit platform. The baseline survey captured sociodemographic characteristics and key indicators across the maternal and newborn health continuum, as well as selected sexual and reproductive health measures. Follow-up interviews were conducted at 6 weeks, 6 months, and one year postpartum. All interviews were administered face-to-face by trained female resident enumerators in local languages. Enumerators received comprehensive training on ethical research conduct, digital data collection tools, and survey procedures [22]. The six-week postpartum follow-up took place between November 2021 and November 2022, collecting comprehensive information on antenatal care, delivery, immediate postpartum, and neonatal care. Baseline interviews captured sociodemographic and household characteristics to enable linkage with the follow-up data [21].

Trained female enumerators conducted face-to-face interviews in local languages after undergoing comprehensive training on ethical conduct, digital data tools, and survey procedures. To ensure high data quality, the survey implemented rigorous protocols including real-time field supervision, standardized training, random spot-checks, and back-checking of key responses. The use of automated skip logic and built-in consistency checks within the ODK platform helped minimize data entry errors. Only panel-eligible women completed the full female questionnaire, and data were structured in wide format to facilitate longitudinal analysis. Sampling weights were applied to correct for design effects and non-response, ensuring nationally representative and reliable estimates [21].

### Data extraction procedure

A total of 2,297 women participated in the baseline interview and gave their consent to be followed up at six weeks. Of these, 2,024 women (1,796 currently pregnant and 228 in the early postpartum period) were eligible for follow-up. Among them, 1,799 completed the six-week interview, while 225 were lost to follow-up due to reasons such as relocation, refusal, or ineligibility. Additionally, 273 women who were already 5–9 weeks postpartum at baseline completed the six-week interview concurrently. All 2,072 women who took part in the follow-up interviews delivered live-born infants. From these, 1,660 postpartum women (with 1,685 infants including 25 twin births) and the 273 postpartum women (with 275 infants including 2 twin births) were assessed for any infant illness since birth. In total, 1,960 infants were included in the final analysis [21] (Fig 1).

### Variables

**Dependent Variable. Presence of any illness since birth, defined as a binary outcome (ill/not ill).**
**Independent Variables.** Individual level factors
**Sociodemographic variables**: The sociodemographic characteristics considered in the study include maternal age, the gender of the newborn, religion, marital status, and the household wealth index, the number of live births (parity), household food insecurity status were assessed in this study.
**Maternal and pregnancy-related factors:** The number of antenatal care visit and timing of first antenatal care visits, whether the mother experienced intimate partner violence. Additional variables included pregnancy reaction of mother, pregnancy reaction of partner or husband, pregnancy type (singleton vs twin), the extent of respectful maternity care received, and whether the mother received birth preparedness and complication readiness (BPCR) counseling,

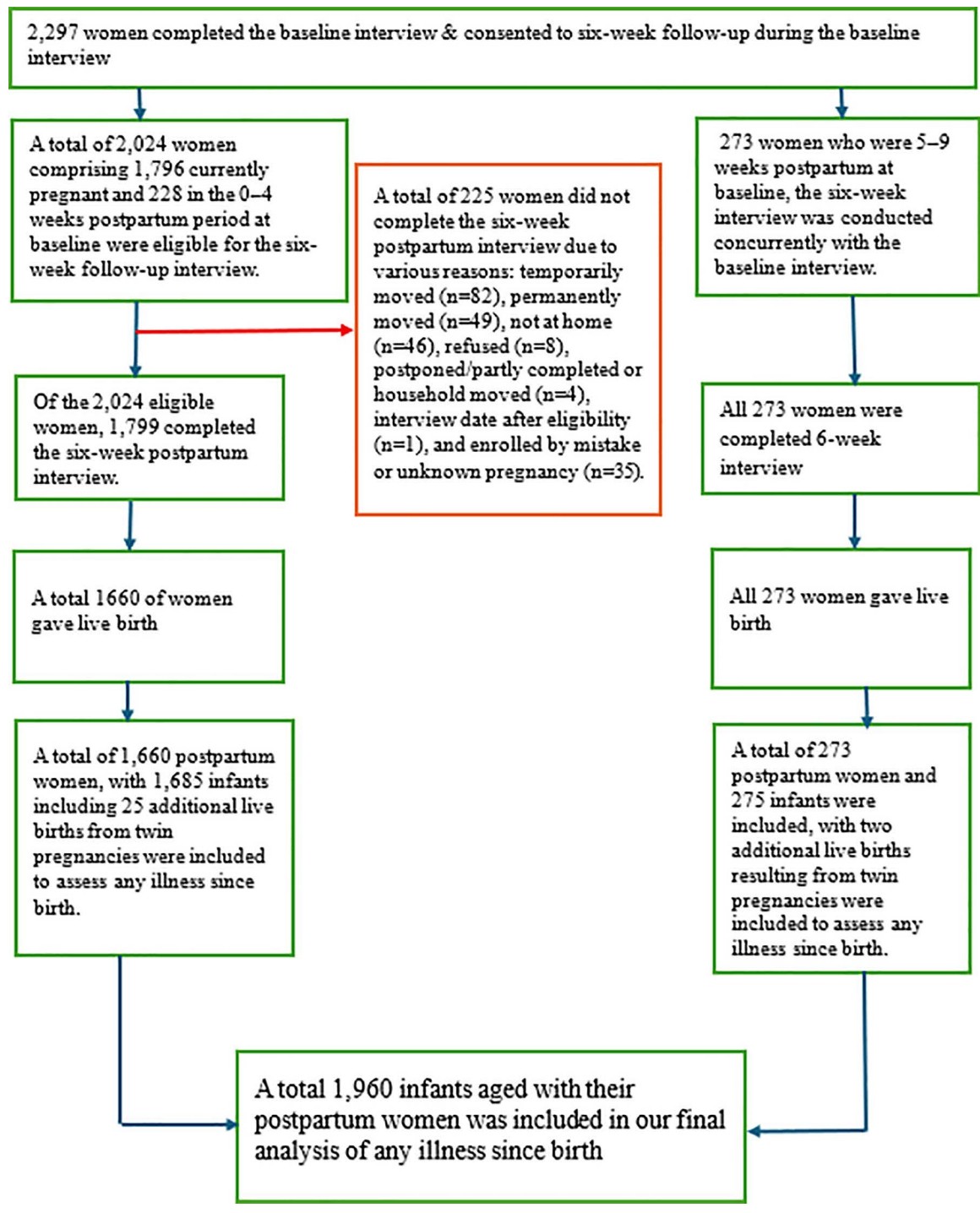

**Fig 1. Data extraction procedure for assessing multilevel determinants of illness since birth in Ethiopia: evidence from Cohort 2 of the longitudinal survey (2021–2023).**

occurrence of pregnancy complication, presence of delivery complications, the place of deliver, cesarean section delivery, the presence of skilled birth attendance, and the occurrence of post-delivery complications, whether the baby was wrapped immediately after birth, appropriate cord care (substances used, instruments used), and early or delayed bathing practices, initiation of breast feeding within 1 hour birth, baby placed naked on mother's chest immediately after birth tetanus immunization during pregnancy, ANC iron supplementation given during her pregnancy, ANC deworming supplementation received during pregnancy were included as maternal and pregnancy related factors.

Community Level factors: Community-level factors included the place of residence, region, community's education status, and community's poverty status.

**Operational definition. Illness:** Refers to any health condition reported by the mother that affected the infant from birth up to six weeks of age. This includes, but is not limited to, symptoms such as vomiting, unconsciousness, sore throat, umbilical redness or pus, poor feeding, absence of urine or stool, failure to cry or breathe, lethargy, skin rash or lesions, jaundice, hypothermia, fever, fast breathing, red or pus-filled eyes, difficulty breathing, diarrhea, cold or cough, convulsions, or chest indrawing. An infant is classified as "**ill**" if they exhibit any one of these symptoms. **Not Ill:** Infants who did not show any of the above symptoms or conditions during the first six weeks of life are classified as "not ill."

**Wealth quintile**: Household wealth status was measured using the wealth quintile variable provided by the PMA-Ethiopia project. PMA constructs this variable using an asset-based wealth index derived through principal component analysis (PCA) of household ownership of durable assets, housing characteristics, and access to water and sanitation facilities. The resulting wealth score is ranked at the national level and divided into five quintiles such as poorest, poorer, middle, richer, and richest and each representing 20% of households. This study used the PMA-constructed wealth quintile variable as provided in the dataset [22].

**Household food insecurity status:** Household food insecurity was assessed using nine questions and follow-up items on the frequency of food insufficiency and related psychological responses over the past 30 days [23,24].

**Delivery Complications**: All participants were asked whether they encountered any issues during childbirth. Reported complications included heavy bleeding, prolonged rupture or leakage of membranes without the onset of labor for over 24 hours, premature membrane rupture before nine months of gestation, abnormal fetal position or presentation, labor lasting more than 12 hours, and episodes of convulsions or seizures. Affirmative response to any complication was categorized into a variable for "any complication." All complications were self-reported and were not confirmed via record review or formal diagnosis.

**Birth preparedness discussion and complication readiness:** women with ANC who received counseling on each birth preparedness topic, including place of delivery, delivery by a skilled birth attendant, arrangement of delivery transport, where to go when experiencing pregnancy danger signs, severe headaches with blurred vision, high blood pressure, edema, convulsions, and bleeding before delivery as a danger sign [21].

**Community poverty and community education level:** Community-level indicators of poverty and education were derived by aggregating individual-level characteristics within each community. These indicators were categorized into "high" or "low" based on the proportion of relevant attributes observed in each community cluster. Specifically, community poverty was classified as high if the proportion of households in the lowest two wealth quintiles (lowest and lower) within a cluster exceeded the median value across all clusters; otherwise, it was considered low. Likewise, the community literacy level was categorized as high if the percentage of women with a minimum of primary education including primary, secondary, technical, vocational, or higher education, which was equal to or greater than the median value within the cluster; it was considered low if it was below the median. This median-based dichotomization accounted for the uneven distribution of poverty and literacy across communities and was also applied to create community-level variables related to the presence of any illness since birth.

## Data management and analysis

The dataset, which is publicly accessible through the PMA website, was downloaded in Stata format and analyzed using Stata version 17. Data preparation involved extracting the study variables and extracting our study

population, labeling variables, recoding categorical responses, generating composite indicators, and addressing missing values. The dataset was restructured to reflect its multilevel hierarchy: infants (level 1) nested within mothers (level 2), who were further nested within enumeration areas (EAs) (level 3). The outcome variable, infant illness, was defined as a binary variable indicating whether the infant experienced any reported symptoms since birth. The independent variables were grouped into two categories: individual-level factors and community-level determinants. Given the complex, stratified multistage sampling design of the PMA survey, all analyses incorporated sampling weights, cluster identifiers, and strata to ensure representativeness and adjust for design effects and nonresponse [25].

To account for the hierarchical nature of the data, multilevel binary logistic regression was employed. Initially, bivariable analyses were performed, and variables with a p-value < 0.25 were selected for inclusion in the multivariable model. After determining the key variables influencing the outcome, four models were developed sequentially: Model 0 (the null model) assessed the initial variation across clusters; Model 1 included individual-level factors; Model 2 introduced community-level variables; and Model 3 integrated predictors from both individual and community levels. Multicollinearity was assessed using the Variance Inflation Factor (VIF), with variables exceeding a VIF of 10 either removed or conceptually merged. Model fit was evaluated using Akaike Information Criterion (AIC) and Bayesian Information Criterion (BIC), with lower values indicating better fit [26]. The degree of clustering was assessed through the Intraclass Correlation Coefficient (ICC) and Median Odds Ratio (MOR) [27], while the Proportional Change in Variance (PCV) [28] quantified variance explained by predictors across models. Fixed-effect estimates were reported as adjusted odds ratios (AORs) with 95% confidence intervals (CIs), and statistical significance was determined at p < 0.05. Results were presented in well-structured texts, tables and Figs, progressing from descriptive summaries to inferential multilevel analyses, with interpretation highlighting both statistical and public health relevance in the context of maternal and child health in Ethiopia.

## Model formulation for multilevel binary logistic regression analysis

Logistic regression uses the logit transformation to convert the non-linear relationship between predictor X and the probability of outcome Y into a linear form. This is achieved by using odds and logarithms, with the logit function representing the S-shaped curve. Generalized linear models refer to a class of models that uses a relation function to make an estimation. The logit link function is used in binary logistic regression [29–31]. Let $X_{ijk} = (X_{1ijk}, X_{2ijk} \dots, X_{nijk})'$ represent the vector of k predictor variables for the $i^{th}$ individual in $j^{th}$ postpartum mother in $k^{th}$ enumeration area, where i = 1, 2…,…, $i^{th}$ infant; j = 1,2,3, 4…, $j^{th}$ postpartum mother; k = 1,2,3, 4…, $k^{th}$ enumeration area)

The probability of any illness since birth in $i^{th}$ infant in $j^{th}$ postpartum mother, in the $k^{th}$ enumeration area and with their predictors $X_{ijk}$ is given by:

$$p_{ij} = \text{prob}\,(y_{ijk} = 1|X_{ijk}) = \frac{ex'_{ijk}\beta}{1 + ex'_{ijk\beta}} = \frac{e^{\beta_0 + x_{1ijk}\beta_1 + x_{2ijk}\beta_2 + \dots, + x_{nijk}\beta_n}}{1 + e^{\beta_0 + x_{1ijk}\beta_1 + x_{2ijk}\beta_2 + \dots, + x_{nijk}\beta_n}}$$

Where β = (β₀, β₁, β₂, …, β) ′ represents a denotes a vector consisting of unknown parameters from the population.

$$
Y_{ijk} = \begin{cases}
1, & \text{if the } i^{th} \text{ infant in } j^{th} \text{ postpartum mother } in\ the\ k^{th} \text{ enumeration area} \\
 & \text{had any types of illnes since birth} \\
0, & \text{if the } i^{th} \text{ infant in } j^{th} \text{ postpartum mother } in\ the\ k^{th} \text{ enumeration area} \\
 & \text{had no any types of illnes since birth}
\end{cases}
$$

The logit transformation of $p_{ijk}$ is expressed as a linear function of the explanatory variables.

$$\text{logit } (p_{ijk}) = \ln\left[\frac{p_{ijk}}{1 - p_{ijk}}\right] = X'_{ijk}\beta = \beta_0 + x_{1ijk}\beta_1 + x_{2ijk}\beta_2 + , \ldots , x_{nijk}\beta_n$$

## Measures of clustering and between-level heterogeneity

### 1. Intraclass correlation coefficient

This study used a three-level binary logistic regression model, where the levels are: Level 1: infants' level (outcome: illness since birth), Level 2: Postpartum mother (mother-level clustering), and Level 3: Enumeration Area ID (community-level clustering),

Level 1 residual variance (for logistic models): For logistic models, the individual-level residual variance is fixed at $\sigma^2 residual = \frac{\pi^2}{3} \cong 3.29$ [32]. The following formulas outline how to compute the ICC at both the postpartum mother level and the enumeration area level, as well as the combined clustering effect.

i. ICC at the Postpartum Mother Level

$$\text{ICC for postpartum clustering level} = \frac{\sigma^2 postpartum\ level}{\sigma^2 enumeration\ area + \sigma^2 postpartum\ women\ level + \sigma^2 residual}$$

This measures the proportion of total variance in infant illness attributable to differences between postpartum women within the same enumeration area

ii. ICC at the Enumeration Area (EA) Level

$$\text{ICC for enumeration area} = \frac{\sigma^2 enumeration\ area}{\sigma^2 enumeration\ area + \sigma^2 postpartum\ women\ level + \sigma^2 residual}$$

It indicates the variation in any illness since birth due to differences between enumeration areas. It tells proportion of the residual variance in the likelihood of infants illness that is attributable by the differences between in enumeration areas [32,33].

iii. Combined Clustering Effect (EA + Postpartum women level)

$$\text{ICC Combined} = \frac{\sigma^2 enumeration\ area + \sigma^2 postpartum\ women\ level}{\sigma^2 enumeration\ area + \sigma^2 postpartum\ women\ level + \sigma^2 residual}$$

This indicates the overall explained variation by clustering effect of both postpartum mother level and enumeration area level. The remainder was attributed to unexplained residual variation at the individual level, as modeled by the logistic distribution. It tells the residual variance is attributable to differences between individuals (postpartum mother level ID) within the same Enumeration area cluster.

### 2. Median odds ratio

The median odd ratio (MOR) quantifies the unexplained cluster-level (random effects) heterogeneity on the odds ratio scale and represents the median increase in the odds of the outcome when comparing two randomly selected individuals from different clusters (postpartum mother or enumeration area), attributable solely to clustering. The MOR translates the

random effects (variances) into the odds ratio scale. the MOR quantifies how much cluster-level differences contribute to the likelihood of the outcome, offering a more intuitive interpretation of random effects than raw variance estimates [27,34].

To compute the Median Odds Ratio in a three-level binary logistic regression model, where the levels are: Level 1: infant level (outcome: illness since birth), Level 2: Postpartum mother ID (mother-level clustering), and Level 3: Enumeration Area ID (community-level clustering), For a given level (say level j), the MOR is calculated as:

$$MOR_{j^{th}} = e^{\left(\sqrt{2*\sigma_j^2}*\phi^{-1}(0.75)\right)}$$

Where; $\sigma_j^2 ==$ estimated variance at level $j^{th}$ cluster (Enumeration Area or Mother)

$\phi^{-1}(0.75)= 0.6745 =$ estimated variance at level $j^{th}$ cluster (the clusters considered in this study were Enumeration Area or postpartum Mother level)

i.  MOR at enumeration area

$$MOR_{j^{th}\ enumeration\ area} = e^{\left(\sqrt{2*\sigma_{j^{th}\ enumeration\ area}^2}*\phi^{-1}(0.75)\right)}$$

This indicates the median increase in odds of an illness for two infants from different enumeration areas *purely due to differences between EAs.*

ii.  MOR at postpartum mother level

$$MOR_{j^{th}\ postpartum\ mother} = e^{\left(\sqrt{2*\sigma_{j^{th}\ postpartum\ mother\ level}^2}*\phi^{-1}(0.75)\right)}$$

MOR at postpartum mother level tells by how much the median increase in odds of illness since birth for two infants from different postpartum mothers *purely due to differences between postpartum mothers*. This shows a substantial variation at the mother level.

iii.  The Overall Median Odds Ratio (MOR combined)

This tells you the median odds ratio between two individuals from different enumeration area and different postpartum mothers, purely due to random effects at both levels. It tells due to unmeasured differences at both the enumeration area and postpartum mother level. The median odds of illness between two randomly selected individuals from different postpartum mothers in different enumeration area level can computed as follows:

$$Overall\ MOR = e^{\left(\sqrt{2*(\sigma_{j^{th}\ enumeration\ area}^2 + \sigma_{j^{th}\ postpartum\ mother}^2)}*\phi^{-1}(0.75)\right)}$$

3.  **Proportion Change in Variance**

The Proportional Change in Variance (PCV) quantifies the proportion of variance at a specific level that is explained after additional predictors are included in the model [28].

$$PCV = \frac{Variance\ at\ null\ model - Variance\ \ of\ adjacent\ model}{variance\ at\ null\ model}$$

i. Proportional change in variance at enumeration area level

$$PCV \text{ at enumeration area level} = \frac{Variance \text{ at null model} - Variance \text{ at adjacent model}}{variance \text{ at null model}}$$

This reflects the reduction in unexplained variation at the enumeration area level after accounting for additional covariates in the model.

ii. Proportion changes in variance at postpartum mother level

$$PCV \text{ at postpartum mother level} = \frac{Variance \text{ at null model} - Variance \text{ at adjacent model}}{variance \text{ at null model}}$$

It measures the extent to which added predictors reduce the unexplained variance between postpartum women.

iii. Proportional change in variance combined (both at enumeration area + at postpartum mother level)

$$PCC \text{ Combined} = \frac{\begin{array}{c}(\sigma^2 \text{ enumeration area null model} + \sigma^2 postpartum \text{ women level at null model}) \\ - (\sigma^2 enumeration \text{ area of adjacent model} + \sigma^2 postpartumwomen \text{ level})\end{array}}{\sigma^2 enumeration \text{ area null model} + \sigma^2 postpartum \text{ women level of the null model}}$$

This combined PCV quantifies the overall proportion of variation explained by predictors at both the enumeration area and postpartum mother levels, highlighting the impact of model adjustments on reducing unexplained heterogeneity.

**Ethical considerations and approval**

This dataset can be accessed freely via their official platform (https://www.pmadata.org/dataset). All analyses adhered to ethical standards and protocols approved by the Institutional Review Boards (IRBs) governing the original PMA data collection, in accordance with the principles of the **Declaration of Helsinki**. The dataset contained no personal identifiers, and strict confidentiality measures were applied to protect the privacy of respondents, households, and sampled communities. Since this study is based on secondary data analysis of publicly available anonymized data, further ethical approval and participant consent were not required. Additional details regarding data ethics and protection measures can be accessed through the PMA-Ethiopia website.

## Results

**Sociodemographic characteristics of respondents**

This study included a total of 1,960 women from the Ethiopia PMA Cohort 2 longitudinal survey (2021–2023). This summary presents the sociodemographic characteristics of infants' mother based on weighted percentages. The majority of respondents were aged 25–29 years (29.1%), followed by those aged 20–24 years (24.5%). Most participants resided in rural areas (75.4%) and were currently married or living with a partner (97.5%). In terms of educational attainment, nearly one-third (31.2%) of mothers had never attended school, while 46.4% had completed primary education. A small proportion (4.8%) attained higher education. Regarding religious affiliation, Muslim (36.0%) and Orthodox Christian (33.1%) faiths were the most commonly reported. Regionally, over half (52.4%) of the participants were from Oromia, with others from SNNPR (21.9%), Amhara (21.1%), and Addis Ababa (4.7%). Newborns were almost equally distributed by sex, with 50.2% male and 49.8% female. Household food security was a challenge, with 31.0% experiencing moderate food insecurity and 13.7% facing severe food insecurity. The wealth distribution was fairly even across quintiles, ranging from 19.3%

(poorest) to 20.3% (richest). The majority of mothers (80.5%) had given birth previously (multiparous), whereas 11.7% were first-time mothers (nulliparous). At the community level, 71.9% lived in areas with high educational status, and 95.4% were from communities with low poverty levels (Table 1).

**Maternal and newborn care related characteristics of respondents**

This analysis summarizes maternal and newborn care practices among infants' mothers, based on nationally representative weighted data. A substantial proportion of mothers (27.8%) did not receive any antenatal care (ANC), while 39.8% attended 1–3 visits, and 32.5% had four or more. In terms of the timing of ANC initiation, 34.0% began within the first four months of pregnancy, 38.3% initiated between five and nine months, and 27.8% received no ANC at all. Regarding health education and preventive measures during pregnancy, 62.1% of mothers reported not discussing nutrition or diet during ANC visits. Although 76.3% received a tetanus injection, only 15.9% consumed multivitamins containing iron, and the same percentage took medication for intestinal worm infections. In terms of service quality, 79.5% of mothers had their weight measured as part of ANC, while 20.5% did not. Almost all pregnancies (97.1%) were singletons, with twin pregnancies accounting for only 2.9%. Emotional reactions to pregnancy varied, with 41.3% of mothers feeling very happy and 32.6% somewhat happy, whereas 5.4% reported being very unhappy. Partners generally responded more positively, with 49.8% being very happy and just 1.5% expressing strong dissatisfaction. Intimate partner violence during pregnancy was reported by 10.7% of mothers. Notably, just 6.5% of mothers were adequately prepared for childbirth and possible complications, highlighting a critical gap in birth preparedness (Table 2).

In terms of delivery settings, 37.9% of births occurred at home, while 59.9% took place in government health facilities. A small proportion delivered in private (1.2%) or NGO/faith-based facilities (0.5%). Overall, 62.1% of deliveries occurred in health facilities. Skilled attendants assisted 61.7% of mothers during delivery, whereas 38.3% were assisted by unskilled personnel. Cesarean sections were performed in 5.7% of births. Maternal health complications were common: 43.5% experienced complication during pregnancy, 33.4% encountered complications during delivery, and 29.0% reported postnatal complications. Despite these risks, respectful maternity care was notably low, only 4.6% of mothers received such care, while the majority (95.4%) did not (Table 2).

Immediate newborn care practices showed mixed adherence to recommended standards. Most newborns (95.5%) were wrapped immediately after birth, but only 52.8% experienced skin-to-skin contact with their mothers. In 75.8% of cases, appropriate substances were used for cord care, while the remaining 24.3% involved the application of unsuitable materials. Delayed bathing, a protective newborn practice, was observed in 59.0% of cases. However, early initiation of breastfeeding within one hour of birth occurred in just 15.5% of cases, indicating a significant gap in optimal neonatal care practices (Table 2).

**Environmental health and sanitation related characteristics of respondents**

The findings reveal environmental health and sanitation related characteristics among the study population. A significant proportion of households (78.2%) did not have access to insecticide-treated bed nets. About 84.1% of households using unimproved toilet facilities and only 15.9% accessing improved toilet facilities. Cooking fuel uses further reflected environmental risks, as 86.4% relied on moderately polluting fuels and 6.5% on highly polluting ones, while just 7.1% used clean fuels. Although 72.6% of households had access to improved drinking water sources, 27.4% still relied on unimproved water sources (Table 3).

**weighted prevalence of infant illness in the first six weeks of life**

Based on weighted data from 1,960 infants aged 5–9weeks, about 34.42% (95% CI: 30.23, 38.87) of infants experienced at least one illness episode since illness. In contrast, 65.58% (95% CI: 61.13, 69.77) of infants remained illness-free during this early infancy period.

**Table 1. Sociodemographic-economic related characteristics of respondents for the assessments of multilevel determinants of illness since birth in Ethiopia: Evidence on Cohort 2 PMA-Ethiopia Longitudinal survey (2021-2023).**

| Variables | Categories | Unweighted | | Weighted | |
|---|---|---|---|---|---|
| | | Frequency | Percentage | Frequency | Percentage |
| Age of the mother in years | 15-19 | 191 | 9.74 | 234 | 11.94 |
| | 20-24 | 472 | 24.08 | 480 | 24.53 |
| | 25-29 | 607 | 30.97 | 570 | 29.12 |
| | 30-34 | 385 | 19.64 | 377 | 19.27 |
| | 35-49 | 305 | 15.56 | 296 | 15.13 |
| Residence | Urban | 822 | 41.94 | 482 | 24.65 |
| | Rural | 1138 | 58.06 | 1,475 | 75.35 |
| Gender of newborn | Male | 994 | 50.71 | 983 | 50.21 |
| | Female | 966 | 49.29 | 974 | 49.79 |
| Religion | Orthodox | 761 | 38.83 | 647 | 33.05 |
| | Protestant | 611 | 31.17 | 570 | 29.13 |
| | Moslem/Muslim | 564 | 28.78 | 704 | 35.99 |
| | Others (No religion/ non-believer or catholic | 24 | 1.22 | 36 | 1.83 |
| Marital status | Currently married/living with man | 1,908 | 97.35 | 1,908 | 97.49 |
| | Divorced/separated/not currently/single | 52 | 2.65 | 49 | 2.51 |
| Region | Amhara | 425 | 21.68 | 412 | 21.05 |
| | Oromia | 701 | 35.77 | 1,026 | 52.40 |
| | SNNPR | 560 | 28.57 | 428 | 21.88 |
| | Addis | 274 | 13.98 | 91 | 4.67 |
| School | Never attended | 528 | 26.94 | 610 | 31.16 |
| | Primary | 861 | 43.93 | 909 | 46.44 |
| | Secondary | 334 | 17.04 | 285 | 14.54 |
| | Technical & vocational | 83 | 4.23 | 59 | 3.02 |
| | Higher | 154 | 7.86 | 95 | 4.84 |
| Household Food Insecurity Access category | Food secure | 921 | 47.06 | 897 | 45.92 |
| | Mildly Food Insecure Access | 189 | 9.66 | 183 | 9.39 |
| | Moderately Food Insecure Access | 583 | 29.79 | 606 | 31.03 |
| | Severely Food Insecure Access | 264 | 13.49 | 267 | 13.66 |
| Wealth quintile | Poorest | 393 | 15.46 | 378 | 19.31 |
| | Poor | 310 | 15.82 | 400 | 20.43 |
| | Medium | 314 | 16.02 | 386 | 19.73 |
| | Rich | 381 | 19.44 | 396 | 20.24 |
| | Richest | 652 | 33.27 | 397 | 20.29 |
| Numbers of live birth | Null para | 240 | 13.09 | 214 | 11.72 |
| | Multi-para [1–4] | 1290 | 70.34 | 69 | 80.48 |
| | Grand multipara (>= 5) | 304 | 16.58 | 357 | 19.52 |
| Community education status | Low education | 462 | 23.57 | 551 | 28.13 |
| | High education | 1,498 | 76.43 | 1,406 | 71.87 |
| Community poverty status | Low poverty | 1,893 | 96.58 | 1, 867 | 95.38 |
| | High poverty | 67 | 3.42 | 90 | 4.62 |

**Table 2. Maternal and newborn care related characteristics of respondents for the assessments of multilevel determinants of illness since birth in Ethiopia: Evidence on Cohort 2 PMA-Ethiopia Longitudinal survey (2021-2023).**

| Variables | Categories | Unweighted | | Weighted | |
|---|---|---|---|---|---|
| | | Frequency | Percentage | Frequency | Percentage |
| Number of ANC visit | No ANC visit | 526 | 33.98 | 543 | 27.75 |
| | 1-3 | 666 | 39.18 | 778 | 39.76 |
| | >=4 | 768 | 26.84 | 636 | 32.50 |
| First ANC visit | 0-4 months | 751 | 38.32 | 665 | 33.98 |
| | 5-9 months | 683 | 34.85 | 749 | 38.28 |
| | No ANC visit | 526 | 26.84 | 543 | 27.75 |
| Discussed nutrition or diet during ANC | No | 935 | 58.55 | 989 | 62.13 |
| | Yes | 662 | 41.45 | 603 | 37.87 |
| Given tetanus injection during this pregnancy | No | 377 | 23.61 | 377 | 23.66 |
| | Yes | 1,220 | 76.39 | 1,215 | 76.34 |
| Consumed multi vitamins that contains iron during this pregnancy | No | 1657 | 84.54 | 1,645 | 84.08 |
| | Yes | 303 | 15.46 | 311 | 15.92 |
| Consumed any drug for Intestinal worms during this pregnancy | No | 1,657 | 84.54 | 1,645 | 84.08 |
| | Yes | 303 | 15.46 | 311 | 15.92 |
| Weight taken as part of ANC | No | 260 | 16.28 | 326 | 20.48 |
| | Yes | 1,337 | 83.72 | 1, 266 | 79.52 |
| Number of children in this pregnancy | Single | 1904 | 97.14 | 1901 | 97.11 |
| | Twin | 56 | 2.86 | 56 | 2.89 |
| Pregnancy reaction | Very happy | 863 | 44.14 | 807 | 41.33 |
| | Sort of happy | 620 | 31.71 | 636 | 32.58 |
| | Mixed happy & unhappy | 181 | 9.26 | 189 | 9.65 |
| | Sort of unhappy | 189 | 9.67 | 216 | 11.07 |
| | Very unhappy | 102 | 5.22 | 105 | 5.38 |
| Partner pregnancy reaction | Very happy | 863 | 44.14 | 935 | 49.78 |
| | Sort of happy | 620 | 31.71 | 721 | 38.36 |
| | Mixed happy & unhappy | 181 | 9.26 | 112 | 5.94 |
| | Sort of unhappy | 189 | 9.67 | 83 | 4.42 |
| | Very unhappy | 102 | 5.22 | 28 | 1.51 |
| Intimate partner violence | No violence | 1,504 | 89.47 | 1504 | 89.30 |
| | Violence | 177 | 10.53 | 180 | 10.70 |
| Birth preparedness & complication readiness | Yes | 125 | 92.17 | 104 | 6.50 |
| | No | 1472 | 7.83 | 1488 | 93.50 |
| Place of delivery | Home | 574 | 29.29 | 742 | 37.92 |
| | Governmental | 1,310 | 66.84 | 1172 | 59.87 |
| | Private | 44 | 2.24 | 23 | 1.16 |
| | Other | 9 | 0.46 | 10 | 0.52 |
| | NGO/faith-based health facility | 23 | 1.17 | 10 | 0.52 |
| Health facility delivery | Home | 574 | 29.29 | 742 | 37.91 |
| | Health facility | 1,386 | 70.71 | 1,215 | 62.09 |
| Skilled delivery | Not skilled | 577 | 29.45 | 749 | 38.28 |
| | Skilled | 1,382 | 70.55 | 1,207 | 61.72 |
| Delivered by cesarian section | No | 1,775 | 90.56 | 1,846 | 94.34 |
| | Yes | 185 | 9.44 | 111 | 5.66 |

*(Continued)*

**Table 2.** (Continued)

| Variables | Categories | Unweighted | | Weighted | |
|---|---|---|---|---|---|
| | | Frequency | Percentage | Frequency | Percentage |
| Complication during pregnancy | No complication occurred | 1,117 | 57.37 | 1096 | 56.46 |
| | Complication occurred | 830 | 42.63 | 845 | 43.54 |
| Any Delivery complication | No | 1,289 | 65.80 | 1302 | 66.57 |
| | Yes | 670 | 34.20 | 657 | 33.43 |
| Post delivery complication | Not post-delivery complication | 1,403 | 71.88 | 1,384 | 71.02 |
| | At least have one post-delivery complication | 549 | 28.13 | 28.98 | 28.98 |
| Respectful maternity care | Not received respectful maternity care | 1,303 | 94.83 | 1,147 | 95.37 |
| | Received respectful maternity care | 71 | 5.17 | 56 | 4.63 |
| Baby wrapped after birth | No | 91 | 4.65 | 89 | 4.53 |
| | Yes | 1,868 | 95.35 | 1, 867 | 95.47 |
| Baby placed naked on mother's chest immediately | No | 839 | 42.83 | 923 | 47.20 |
| | Yes | 1,120 | 57.17 | 1,032 | 52.80 |
| Applied substances cord care | Inappropriate | 395 | 21.27 | 450 | 24.25 |
| | Appropriate | 1,462 | 78.73 | 1,405 | 75.75 |
| Bathing time since birth | Early bathing | 694 | 35.72 | 794 | 41.03 |
| | Delayed bathing | 1,249 | 64.28 | 1,141 | 58.97 |
| Initiation of breast feeding | Greater than 1hour | 1,623 | 84.40 | 1,622 | 84.54 |
| | Within 1 hour | 300 | 15.60 | 296 | 15.46 |

**Table 3.** Environmental health and sanitation related characteristics of respondents for the assessments of multilevel determinants of illness since birth in Ethiopia: Evidence on Cohort 2 PMA-Ethiopia Longitudinal survey (2021-2023).

| Variables | Categories | Unweighted | | Weighted | |
|---|---|---|---|---|---|
| | | Frequency | Percentage | Frequency | Percentage |
| Household has an insecticide treated bed net | No | 1,507 | 76.89 | 1,530 | 78.16 |
| | Yes | 453 | 23.11 | 427 | 21.84 |
| Types of toilets | Improved | 508 | 25.92 | 311 | 15.87 |
| | Unimproved | 1,452 | 74.08 | 1,646 | 84.13 |
| Cooking-fuel type | Clean fuel | 292 | 14.90 | 138 | 7.07 |
| | Moderately polluting fuel | 1,568 | 80.00 | 1,691 | 86.43 |
| | Highly polluting fuel | 100 | 5.10 | 127 | 6.50 |
| Sources of drinking | Improved | 1,571 | 80.15 | 1, 421 | 72.64 |
| | Unimproved | 389 | 19.85 | 535 | 27.36 |

### The proportion illness on sociodemographic characteristics of respondents

The proportion of infants who experienced at least one illness since birth showed notable variation across maternal sociodemographic characteristics. Infants born to younger mothers aged 15–19 years had a relatively lower illness rate (33.2%) compared to those born to mothers aged 25–29 years (38.8%), while the lowest rate was observed among those aged 35–49 (30.3%). Rural infants experienced slightly higher illness prevalence (34.6%) than their urban counterparts (33.9%). Gender differences were evident, with male infants showing a higher illness rate (38.3%) compared to females

(30.5%). Infants of currently married mothers had a slightly lower illness burden (34.3%) than those of unmarried or separated women (38.7%). Regional disparities were considerable, with Oromia reporting the lowest illness rate (30.3%) and Amhara (39.8%) and SNNPR (38.7%) the highest. Based on maternal education status, as infants of mothers with primary education had a lower illness rate (31.9%) than those whose mothers had no formal education (37.4%). Similarly, infant illness declined with increasing household wealth, from 43.2% in the poorest quintile to 29.3% in middle-income households. Regarding household food insecurity status, severely food-insecure households reported the highest infant illness (40.8%) compared to food-secure ones (31.3%). Moreover, multiparous mothers (1–4 births) having infants with lower illness rates (33.7%) than grand multiparas (36.8%) and nulliparas (37.4%). At the community level, higher maternal education (32.6%) and lower poverty (34.2%) had lower prevalence of illness compared to low education (39.1%) and high poverty settings (39.5%) (Table 4).

### The proportion illness on maternal and newborn care related characteristics of respondents

The proportion of infant illness since birth varied across maternal and newborn care related characteristics. Infants whose mothers had no antenatal care (ANC) visits experienced higher illness rates (37.4%) compared to those with 1–3 visits (34.4%) or four or more visits (32.0%). Early initiation of ANC (0–4 months) was slightly associated with lower illness (35.1%) than late initiation (38.3%). Tetanus vaccination during pregnancy showed a lower illness proportion (32.8%) among vaccinated mothers versus 37.8% among unvaccinated. Similar patterns were seen for weight measurement during ANC (32.9% vs. 37.8%). Infants from twin pregnancies experienced markedly higher illness (47.4%) compared to singletons (34.0%). Infants of mothers who were "very unhappy" or "sort of unhappy" for her pregnancy had higher illness rates (42.9% and 39.8%, respectively) compared to those who were "very happy" (33.9%). Likewise, negative partner reactions were associated with higher infant illness, particularly among "sort of unhappy" (48.7%) and "very unhappy" (40.5%) partners. Infants whose mothers experienced intimate partner violence had a higher illness rate (36.5%) than those who did not (33.4%). This study also found that infants of mother who received birth preparedness and complication readiness during her ANC had a notably lower illness rate (25.5%) than those infants' mother who did not received birth preparedness and complication readiness service (34.6%). While place and type of delivery showed minimal variation, cesarean delivery was linked to higher infant illness (40.2%) versus their counterparts (34.3%). Illness was also more frequent among infants born from their mothers who had pregnancy complications (44.2%), delivery complications (42.1%), and postnatal complications (44.9%), compared to those without complications. Lastly, receipt of respectful maternity care did not show substantial differences in infant illness, with similar rates observed between recipients (35.2%) and non-recipients (34.2%) (Table 5).

### The proportion illness on environmental health and sanitation related characteristics of respondents

Infants from households without insecticide-treated bed nets had a lower illness rate (32.4%) than those with nets (41.7%). Use of unimproved toilets was associated with a slightly higher illness rate (34.8%) compared to improved toilets (32.2%). Households using highly polluting cooking fuels had the highest infant illness (41.7%), followed by moderately polluting fuels (34.0%), while clean fuel use was linked to lower illness (32.5%). Notably, infants in households with unimproved drinking water sources experienced significantly more illness (44.1%) than those with improved sources (30.8%), highlighting the impact of poor sanitation and unsafe water on early childhood health (Table 6).

### Random effect analysis

The null model, without covariates, showed substantial clustering of infant illness at both the enumeration area (EA) and postpartum mother levels, with variances of 3.20 and 8.16 respectively, and a combined intra-cluster correlation (ICC) of 77.5%. The median odds ratios (MOR) indicated strong heterogeneity, particularly at the mother level (MOR = 15.3) and overall (MOR = 24.9). Model 1, which included individual level predictors, substantially reduced

**Table 4. The proportion of illness since birth across different categories of sociodemographic characteristics of respondents for the assessments of multilevel determinants of illness since birth in Ethiopia: Evidence on Cohort 2 PMA-Ethiopia Longitudinal survey (2021-2023).**

| Serial number | Variables | Categories | At least one illness | |
|---|---|---|---|---|
| | | | Frequency | Percentage |
| | Age category | 15-19 | 78 | 33.16 |
| | | 20-24 | 150 | 31.33 |
| | | 25-29 | 221 | 38.76 |
| | | 30-34 | 135 | 35.81 |
| | | 35-49 | 90 | 30.34 |
| | Residence | Urban | 163 | 33.89 |
| | | Rural | 510 | 34.60 |
| | Gender of newborn | Male | 377 | 38.33 |
| | | Female | 297 | 30.48 |
| | Religion | Orthodox | 230 | 35.59 |
| | | Protestant | 205 | 35.91 |
| | | Moslem/Muslim | 230 | 32.66 |
| | | Others (No religion/ non-believer/catholic | 9 | 24.38 |
| | Marital status | Currently married/living with man | 655 | 34.32 |
| | | Divorced/separated/not currently/single | 19 | 38.65 |
| | Region | Amhara | 164 | 39.81 |
| | | Oromia | 310 | 30.25 |
| | | SNNPR | 166 | 38.73 |
| | | Addis | 34 | 36.84 |
| | School | Never attended | 228 | 37.42 |
| | | Primary | 290 | 31.88 |
| | | Secondary | 100 | 35.09 |
| | | Technical & vocational | 23 | 38.20 |
| | | Higher | 33 | 35.18 |
| | Household Food Insecurity Access category | Food secure | 281 | 31.28 |
| | | Mildly Food Insecure Access | 62 | 34.03 |
| | | Moderately Food Insecure Access | 222 | 36.57 |
| | | Severely Food Insecure Access | 109 | 40.76 |
| | Wealth quintile | Poorest | 163 | 43.18 |
| | | Poor | 145 | 36.21 |
| | | Medium | 113 | 29.27 |
| | | Rich | 131 | 33.04 |
| | | Richest | 122 | 30.69 |
| | Numbers of live birth | Null para | 80 | 37.38 |
| | | Multi-para [1–4] | 424 | 33.73 |
| | | Grand multipara (>= 5) | 131 | 36.82 |
| | Community education status | Low education | 215 | 39.10 |
| | | High education | 458 | 32.60 |
| | Community poverty status | Low poverty | 638 | 34.18 |
| | | High poverty | 35.7 | 39.50 |

**Table 5. The proportion of illness since birth across different categories of maternal and newborn care related characteristics of respondents for the assessments of multilevel determinants of illness since birth in Ethiopia: Evidence on Cohort 2 PMA-Ethiopia Longitudinal survey (2021-2023).**

| Variables | Categories | At least one illness | |
|---|---|---|---|
| | | **Frequency** | **Percentage** |
| Number of ANC visit | No ANC visit | 203 | 37.39 |
| | 1-3 | 267 | 34.38 |
| | >=4 | 203 | 31.95 |
| First ANC visit | 0-4 months | 233 | 35.08 |
| | 5-9 months | 340 | 62.61 |
| | No ANC visit | 674 | 34.42 |
| Discussed nutrition or diet during ANC | No | 327 | 33.11 |
| | Yes | 213 | 35.35 | |
| Given tetanus injection during this pregnancy | No | 142 | 37.77 |
| | Yes | 398 | 32.78 |
| Consumed multi vitamins that contains iron during this pregnancy | No | 187 | 35.14 |
| | Yes | 487 | 34.16 |
| Consumed any drug for Intestinal worms during this pregnancy | No | 558 | 33.89 |
| | Yes | 116 | 37.26 |
| Weight taken as part of ANC | No | 123 | 37.81 |
| | Yes | 417 | 32.97 |
| Number of children in this pregnancy | Single | 647 | 34.04 |
| | twin | 27 | 47.40 |
| Pregnancy reaction | Very happy | 274 | 33.95 |
| | Sort of happy | 200 | 31.39 |
| | Mixed happy & unhappy | 69 | 36.50 |
| | Sort of unhappy | 86 | 39.75 |
| | Very unhappy | 45 | 42.91 |
| Partner pregnancy reaction | Very happy | 313 | 33.51 |
| | Sort of happy | 243 | 33.79 |
| | Mixed happy & unhappy | 37 | 32.86 |
| | Sort of unhappy | 40 | 48.65 |
| | Very unhappy | 34 | 40.45 |
| Intimate partner violence | No violence | 503 | 33.44 |
| | Violence | 66 | 36.50 |
| Birth preparedness & complication readiness | No | 514 | 34.55 |
| | Yes | 26 | 25.49 |
| Place of delivery | Home | 255 | 34.41 |
| | governmental | 14 | 60.19 |
| | Private | 9 | 39.81 |
| | Other | 5 | 50 |
| | NGO/faith-based health facility | 4 | 39.88 |
| Health facility delivery | Home | 255 | 34.41 |
| | Health facility | 1283 | 65.58 |
| Skilled delivery | Not skilled | 258 | 34.49 |
| | Skilled | 415 | 34.41 |
| Delivered by cesarian section | No | 65.92 | 629 |
| | Yes | 45 | 40.18 |

*(Continued)*

**Table 5.** (Continued)

| Variables | Categories | At least one illness | |
|---|---|---|---|
| | | **Frequency** | **Percentage** |
| Complication during pregnancy | No complication occurred | 294 | 26.81 |
| | Complication occurred | 373 | 44.16 |
| Any Delivery complication | No | 399 | 30.61 |
| | Yes | 275 | 42.10 |
| Post delivery complication | Not post-delivery complication | 418 | 30.20 |
| | At least have one post-delivery complication | 253 | 44.85 |
| Respectful maternity care | Not received respectful maternity care | 392 | 34.17 |
| | Received respectful maternity care | 20 | 35.23 |

**Table 6. The proportion of illness since birth across different categories environmental health and sanitation related characteristics of respondents for the assessments of multilevel determinants of illness since birth in Ethiopia: Evidence on Cohort 2 PMA-Ethiopia Longitudinal survey (2021-2023).**

| Variables | Categories | At least one illness | |
|---|---|---|---|
| | | **Frequency** | **Percentage** |
| Household has an insecticide treated bed net | No | 496 | 32.39 |
| | Yes | 178 | 41.69 |
| Types of toilets | Improved | 100 | 32.22 |
| | Unimproved | 574 | 34.84 |
| Cooking-fuel type | Clean fuel | 45 | 32.53 |
| | Moderately polluting fuel | 376 | 34.03 |
| | Highly polluting fuel | 53 | 41.67 |
| Sources of drinking | Improved | 438 | 30.81 |
| | Unimproved | 236 | 44.08 |

variance at both EA (to 1.09) and mother levels (to 2.35), lowering combined clustering (ICC) to 51.1%. This model explained about 69.7% of the total variance (PCV), indicating improved fit, supported by lower AIC and BIC values. Model 2, incorporating community level predictor, showed higher variances similar to the null model (EA variance 2.98, mother variance 8.18), with clustering effects close to the null model (combined ICC 77.2%) and minimal variance reduction (PCV around 1.7%), suggesting limited explanatory power of these variables alone. Model 3, the full model with all covariates, again reduced variance and clustering substantially (EA variance 1.03, mother variance 2.52, combined ICC 52.0%), with PCV around 69%, similar to Model 1. Median odds ratios reflected reduced heterogeneity (MOR around 2.6 for EA and 4.5 for mother level).

Overall, the analysis indicates that a large portion of variation in infant illness is explained by individual- and community-level factors, with significant clustering at both enumeration area and maternal levels. Sociodemographic and maternal characteristics substantially reduce unexplained variance, whereas community level factors alone contribute less. The final full model (Model 3) was identified as the optimal model based on a comprehensive evaluation of model fit indices and variance components. Considering key metrics such as AIC, BIC, deviance, proportional change in variance (PCV), intra-cluster correlations (ICCs), and median odds ratios (MORs), Model 3 demonstrated superior performance. It had the lowest AIC (1258.30) and deviance (591.15), the highest PCV (68.69%), the lowest combined ICC (51.95%), and a reduced MOR (6.04). These results indicate that Model 3 offers the best fit, explains

the greatest amount of variance, and effectively minimizes unexplained heterogeneity compared to the null model and other candidate models (Table 7).

## Bivariable analysis and multivariable analysis of predictors and presence of any illness since birth

Based on the bivariable analysis, variables with a p-value ≤ 0.25 that were included in the final multivariable multilevel binary logistic regression model comprised maternal age, gender of the newborn, religion, and region; wealth quintile; number of antenatal care visits; receiving tetanus injection during pregnancy; consuming medication for intestinal worms during ANC; and having weight measured as part of ANC. Additionally, factors such as the number of fetuses in the current

**Table 7. Random effect analysis and model fit statistics for the assessments of multilevel determinants of illness since birth in Ethiopia: Evidence on Cohort 2 PMA-Ethiopia Longitudinal survey (2021-2023).**

| Parameter | Null model | Model 1 | Model 2 | Model 3 |
|---|---|---|---|---|
| **Random Effects Variance (95% CI)** | | | | |
| Enumeration area level variance | 3.196(95%CI: 1.409, 7.249) | 1.088 (95% CI:.485, 2.444) | 2.980(95% CI: 1.313, 6.767) | 1.032(95%CI:.453, 2.351) |
| Postpartum mother level variance within enumeration area | 8.161(95%CI:3.194, 20.854) | 2.353 (95% CI:.615, 9.001) | 8.182(95% CI: 3.250, 20.597) | 2.524(95% CI:.686, 9.293) |
| Total cluster level variance | 11.357(95%CI: 4.603, 28.103) | 3.441(95% CI: 1.100, 11.445) | 11.162(95% CI: 4.563, 27.363) | 3.556(95%CI: 1.139,11.644) |
| **Intracluster Correlation Coefficient (ICC, %)** | | | | |
| ICC at enumeration area level (%) | 21.82% (95% CI: 16.18, 28.75) | 16.17% (95% CI: 10.05, 24.98) | 20.62% (95%:15.28, 27.24) | 15.08% (95%CI:9.26, 23.59) |
| ICC at postpartum mother level (%) | 55.72% (95% CI:23.26, 81.64) | 34.96% (95%, 12.32, 67.24) | 56.62% (95% CI: 34.15, 76.66) | 36.87%(95%CI: 13.68, 68.26) |
| Combined Clustering Effect (EA + Postpartum mother level) (%) | 77.54% (95% CI: 58.72, 89.34) | 51.13% (95% CI: 25.45, 76.22) | 77.24% (95%CI:58.48, 89.10) | 51.95% (95% CI: 26.19, 76.71) |
| **Median Odds Ratio (MOR, 95% CI)** | | | | |
| MOR at enumeration area level | 5.50 (95%CI:3.10, 13.04) | 2.71(95% CI: 1.94, 4.44) | 5.19(95%:2.98, 11.96) | 2.64(95% CI: 1.90, 4.32) |
| MOR at postpartum mother level | 15.26(95% CI: 5.50, 77.95) | 4.32(95% CI: 2.11, 17.49) | 15.31(95%CI: 5.58, 75.87) | 4.55(95% CI:2.20, 18.32) |
| Overall MOR (EA + Postpartum women level) | 24.89(95% CI: 7.74, 157.08) | 5.87(95% CI: 2.72, 25.20) | 24.21(95% CI:7.67, 146.90) | 6.04(95%: 2.77, 25.92) |
| **Proportional Change in Variance (PCV, %)** | | | | |
| PCV at enumeration area level | Reference | 65.95% (95%CI: 65.62, 66.28) | 6.75% (95%: 6.85 to 6.65) | 67.71% (95%CI:67.84, 67.57) |
| PCV at postpartum mother level | Reference | 71.17% (95%CI: (80.73, 56.84) | −0.25% (95%CI: −1.75, 1.23) | 69.07% (95%CI: 78.54, 55.44) ` |
| Overall PCV | Reference | 69.70% (95% CI: 76.12,59.27) | 1.72% (95% CI:0.88, 2.63) | 68.69% (95% CI:75.26, 58.57) |
| **Model Fit Indices** | | | | |
| AIC | 2258.929 | 1259.45 | 2258.72 | 1258.30 |
| BIC | 2275.671 | 1435.77 | 2292.21 | 1449.74 |
| Deviance (−2loglikelihood) | 1126.4645 | 594.73 | 1123.36 | 591.15 |

Note that: AIC = Akaike Information Criterion, BIC = Bayesian Information Criterion, CI = Confidence Interval, ICC = Intracluster Correlation Coefficient, MOR = Median Odds Ratio, PCV = Proportional Change in Variance

pregnancy, the mother's emotional response to the pregnancy, the partner's reaction, birth preparedness and complication readiness, cesarean delivery, and the presence of complications during pregnancy, childbirth, and the postpartum period were all taken into account. Furthermore, initiation of breastfeeding within one hour, household use of insecticide-treated bed nets, cooking fuel type, and sources of drinking water were also included in the final model (Table 8).

## Multivariable binary logistic regression analysis

In the final multivariable multilevel binary logistic regression model, several factors were significantly associated with infant illness since birth. Female infants were 47% less likely to experience illness compared to male infants (AOR = 0.53; 95% CI: 0.35, 0.82), suggesting a protective effect of female gender against early-life illnesses. Infants born to mothers who had complications during pregnancy had 89% higher odds of illness (AOR = 1.89; 95% CI: 1.11, 3.20), indicating that maternal health complications during pregnancy substantially increase the risk of infant illness. Moreover, maternal post-delivery complications were associated with more than a twofold increase in the odds of infant illness (AOR = 2.33; 95% CI: 1.27, 4.27), highlighting the critical impact of postpartum maternal health on infant outcomes. Surprisingly, infants from households owning insecticide-treated bed nets showed 81% higher odds of illness (AOR = 1.81; 95% CI: 1.10, 2.96), which may reflect underlying confounding factors such as higher exposure to disease vectors in these households or increased reporting in such settings. Other variables including region, birth preparedness, cesarean delivery, and source of drinking water were not statistically significant in the adjusted model, indicating their limited independent effect on infant illness after controlling for other factors. These findings emphasize the importance of maternal health before and after delivery and gender differences in infant vulnerability, while also suggesting a need to further explore environmental and behavioral factors influencing illness risk (Table 9).

## Discussion

In this study, which utilized weighted data from 1,960 infants aged 5–9 weeks, approximately one in three (34.42%; 95% CI: 30.23–38.87) had experienced at least one episode of illness since birth. This notably high prevalence highlights a substantial burden of early-life morbidity during a critical window when infants are particularly susceptible due to immature immune systems, increased exposure to environmental pathogens, and potential gaps in maternal and newborn care services.

The prevalence found in our study (34.42%) exceeds the 27.8% reported in an earlier study from Northwest Ethiopia [16]. This discrepancy may reflect regional or temporal variations in disease burden, differences in health service utilization, or methodological factors such as differences in sample age, data collection period, illness definition, or weighting approaches. Furthermore, it may suggest a possible increase in early infant morbidity over time, warranting closer monitoring and targeted public health interventions. Comparatively, data from the 2021 Ethiopian Demographic and Health Survey (EDHS) show that among children under five, 21.7% had symptoms of acute respiratory infection (ARI), 12.1% had diarrhea, and 14.8% had fever in the two weeks preceding the survey [3]. While the EDHS Figs refer to slightly older children and a shorter recall period, they still point to a persistently high burden of infectious illnesses among young children in Ethiopia. Our findings, focusing on a younger cohort, emphasize that health vulnerabilities may begin much earlier in infancy. The elevated illness burden observed in this study underscores the pressing need for strengthened postnatal care, early identification of risk factors, and improved maternal health services. These efforts are crucial for addressing preventable infant morbidity and mortality and for accelerating progress toward Sustainable Development Goal (SDG) 3.2, which aims to reduce neonatal mortality to 12 or fewer deaths per 1,000 live births by 2030 [2,35]. Without targeted intervention strategies, Ethiopia may struggle to meet this global benchmark.

Further comparisons with other studies also provide important context. A systematic review and meta-analysis conducted in Ethiopia reported a pooled prevalence of neonatal sepsis at 45% [36], which is higher than the illness prevalence found in our study. Similarly, a hospital-based study in Shashemene, Ethiopia, reported a strikingly high prevalence

**Table 8. Bivariable analysis of each independent variable with presence of any illness since birth for the assessments of multilevel determinants of illness of infants since birth in Ethiopia: Evidence on Cohort 2 PMA-Ethiopia Longitudinal survey (2021-2023).**

| Predictor variables | Categories | Crude odds ratio with its 95% confidence interval | P value |
|---|---|---|---|
| Age of mother in years | 15-19 | Reference | |
| | 20-24 | 1.11(95%: 0.48, 2.54) | 0.81 |
| | 25-29 | 1.85(95%: 0.80, 4.27) | 0.15 |
| | 30-34 | 1.21(95%: 0.49, 3.01) | 0.68 |
| | 35-49 | 0.83(95%: 0.32,2.13 0) | 0.70 |
| Residence | Urban | Reference | |
| | Rural | 0.94(95%:0.46, 1.95) | 0.88 |
| Gender of newborn | Male | Reference | |
| | Female | 0.41(95%: 0.25, 0.65) | <0.001 |
| Religion | Orthodox | Reference | |
| | Protestant | 0.84(95%: 0.46, 1.54) | 0.57 |
| | Moslem/Muslim | 0.86(95%: 0.41, 1.83) | 0.70 |
| | Others (No religion/ non-believer or catholic | 0.35(95%: 0.05, 2.23) | 0.26 |
| Marital status | Currently married/living with man | Reference | |
| | Divorced/separated/not currently/single | 1.38(95%: 0.34, 5.70) | 0.65 |
| Region | Amhara | Reference | |
| | Oromia | 0.34(95%: 0.13, 0.88) | 0.03 |
| | SNNPR | 0.76(95%: 0.26, 2.23) | 0.62 |
| | Addis Ababa | 0.78(95%: 0.28, 2.21) | 0.64 |
| Educational status | Never attended | Reference | |
| | Primary | 0.82(95%: 0.48, 1.43) | 0.49 |
| | Secondary | 1.17(95%: 0.57, 2.38) | 0.67 |
| | Technical & vocational | 1.19(95%: 0.44, 3.22) | 0.73 |
| | Higher | 0.98(95%: 0.41, 2.32) | 0.97 |
| Household Food Insecurity Access category | Food secure | Reference | |
| | Mildly Food Insecure Access | 1.12(95%:0.50, 2.50) | 0.79 |
| | Moderately Food Insecure Access | 1.38(95%:0.72, 2.65) | 0.33 |
| | Severely Food Insecure Access | 1.72(95%: 0.79, 3.77) | 0.17 |
| Wealth quintile | Poorest | Reference | |
| | Poor | 0.72(95%: 0.31, 1.69) | 0.45 |
| | Medium | 0.41(95%: 0.16, 1.05) | 0.06 |
| | Rich | 0.46(95%:0.17,1.27) | 0.14 |
| | Richest | 0.34(95%: 0.12, 0.98) | 0.05 |
| Numbers of live birth | Null para | Reference | |
| | Multi-para [1–4] | 0.82(95%: 0.42, 1.62) | 0.57 |
| | Grand multipara (>= 5) | 1.10(95%:0.48, 2.55) | 0.82 |
| Community education status | Low education | Reference | |
| | High education | 0.87(95%: 0.42, 1.81) | 0.71 |
| Community poverty status | Low poverty | Reference | |
| | High poverty | 1.16(95%: 0.18, 7.58) | 0.87 |
| Number of ANC visit | No ANC visit | Reference | |
| | 1-3 | 0.94(95%: 0.53, 1.68) | 0.84 |
| | >=4 | 0.70(95%: 0.38, 1.31) | 0.27 |

*(Continued)*

**Table 8.** (Continued)

| Predictor variables | Categories | Crude odds ratio with its 95% confidence interval | P value |
|---|---|---|---|
| First ANC visit | 0-4 months | Reference | |
| | 5-9 months | 0.97(95%:0.53, 1.78) | 0.59 |
| | No ANC visit | 1.18(95%:0.65, 2.15) | 0.92 |
| Discussed nutrition or diet during ANC | No | Reference | |
| | Yes | 1.10(95%: 0.63, 1.93) | 0.73 |
| Given tetanus injection during this pregnancy | No | Reference | |
| | Yes | 0.59(95%: 0.33, 1.06) | 0.08 |
| Consumed multi vitamins that contains iron during this pregnancy | No | Reference | |
| | Yes | 0.95(95%: 0.54, 1.66) | 0.86 |
| Consumed any drug for Intestinal worms during this pregnancy | No | Reference | |
| | Yes | 1.47(95%: 0.75, 2.86) | 0.26 |
| Respectful maternity care | Received | Reference | |
| | Not received | 0.92(95%: 0.39, 2.14) | 0.84 |
| Weight taken as part of ANC | No | Reference | |
| | Yes | 0.67(95%: 0.35, 1.29) | 0.23 |
| Number of children in this pregnancy | Single | Reference | |
| | Twin | 4.27(95%: 0.93, 19.49) | 0.06 |
| Pregnancy reaction | Very happy | Reference | |
| | Sort of happy | 0.64(95%: 0.37, 1.14) | 0.13 |
| | Mixed happy & unhappy | 1.49(95%: 0.63, 3.50) | 0.36 |
| | Sort of unhappy | 1.41(95%: 0.60, 3.32) | 0.43 |
| | Very unhappy | 2.97(95%: 1.01, 8.72) | 0.05 |
| Partner pregnancy reaction | Very happy | Reference | |
| | Sort of happy | 1.01(95%: 0.61, 1.66) | 0.98 |
| | Mixed happy & unhappy | 1.22(95%: 0.44, 3.38) | 0.71 |
| | Sort of unhappy | 3.65(95%:1.06, 12.56) | 0.04 |
| | Very unhappy | 0.98(95%: 0.22, 4.33) | 0.98 |
| Intimate partner violence | No | Reference | |
| | Yes | 1.23(95%: 0.50, 3.04) | 0.65 |
| Birth preparedness & complication readiness | No | Reference | |
| | Yes | 0.37(95%: 0.13, 1.02) | 0.05 |
| Health facility delivery | Home | Reference | |
| | Health facility | 1.12(95%: 0.63, 1.99) | 0.69 |
| Skilled delivery | Not skilled | Reference | |
| | Skilled | 1.12(95%: 0.63, 1.96) | 0.71 |
| Delivered by cesarian section | No | Reference | |
| | Yes | 1.65(95%: 0.99, 2.76) | 0.06 |
| Complication during pregnancy | No complication occurred | Reference | |
| | Complication occurred | 4.47(95%: 2.18, 9.19) | <0.001 |
| Any Delivery complication | No | Reference | |
| | Yes | 2.38(95%: 1.33, 4.25) | <0.001 |
| Post delivery complication | No post-delivery complication | Reference | |
| | At least have one post-delivery complication | 3.72(95%: 1.85,7.48) | <0.001 |
| Baby wrapped after birth | No | Reference | |
| | Yes | 0.73(95%:0.24, 2.21) | 0.57 |

*(Continued)*

| Predictor variables | Categories | Crude odds ratio with its 95% confidence interval | P value |
|---|---|---|---|
| Baby placed naked on mother's chest immediately | No | Reference | |
| | Yes | 0.76(95%: 0.47, 1.23) | 0.27 |
| Applied substances cord care | Inappropriate | Reference | |
| | Appropriate | 1.20(95%: 0.71, 2.05) | 0.49 |
| Bathing time since birth | Early bathing | Reference | |
| | Delayed bathing | 0.83(95%: 0.50, 1.38) | 0.47 |
| Initiation of breast feeding | Greater than 1hour | Reference | |
| | Within 1 hour | 1.63(95%: 0.91, 2.92) | 0.10 |
| HH has an insecticide treated bed net | No | Reference | |
| | Yes | 1.86(95%: 1.06,3.26) | 0.03 |
| Types of toilets | Improved | Reference | |
| | Unimproved | 1.27(95%: 0.66, 2.44) | 0.47 |
| Cooking-fuel type used by household | Clean fuel | Reference | |
| | Moderately polluting fuel | 1.51(95%: 0.65, 3.52) | 0.34 |
| | Highly polluting fuel | 2.76(95%: 0.66, 11.49) | 0.16 |
| Sources of drinking | Improved | Reference | |
| | Unimproved | 2.03(95%: 0.98, 4.20) | 0.06 |

of neonatal sepsis at 77.9% [37], likely due to the clinical setting and higher-risk population being studied. On the other hand, our prevalence was higher than that reported in a study using EDHS data, which found the prevalence of ARI among under-five children to be 8.4% [38], again underscoring the increased susceptibility in early infancy.

Additional studies have reported varying rates of neonatal conditions: a study in Bahir Dar, Ethiopia found the prevalence of neonatal jaundice among admitted neonates to be 38.8% [39], while a similar study in Dhaka, Bangladesh reported an even higher rate of 69.35% [40]. In contrast, the prevalence of perinatal asphyxia among newborns delivered at Saad Abuelela Maternity Hospital in Sudan was found to be relatively lower at 11.3% [41]. Collectively, these findings indicate that early infancy remains a period of high vulnerability, with varying prevalence of specific neonatal illnesses across regions and settings. The relatively high burden identified in our study highlights the urgent need for strengthened postnatal care, early screening for neonatal illnesses, and improved maternal health services. Such interventions are critical for reducing preventable morbidity and mortality and achieving Sustainable Development Goal (SDG) 3.2, which targets reducing neonatal mortality to 12 or fewer deaths per 1,000 live births by 2030 [2,35]. Without sustained and targeted public health efforts, Ethiopia may face significant challenges in meeting this global benchmark.

A significant gender disparity was observed in this study, with female infants exhibiting 47% lower odds of illness compared to males (AOR = 0.53; 95% CI: 0.35–0.82). This finding is biologically plausible and consistent with global evidence suggesting that male infants are more vulnerable to early-life illnesses due to several physiological and developmental factors. Biologically, female neonates tend to have more robust immune responses than males, influenced by hormonal and genetic differences. Estrogen enhances both innate and adaptive immunity, while testosterone can have immunosuppressive effects [42]. Furthermore, male infants are at a developmental disadvantage, with slower lung maturation and higher susceptibility to respiratory conditions, sepsis, and other infections [43,44]. These biological distinctions are evident in observed epidemiological trends. Studies have consistently shown higher rates of neonatal and infant morbidity and mortality among males. For example, a large-scale meta-analysis by Mondal et al. (2014) [45] confirmed that male newborns face significantly higher risks of infections and related complications. Similarly, evidence from Ethiopia

**Table 9. Multivariable analysis of each predictor variable with absence or presence of any illness since birth for the assessments of multilevel determinants of any illness since birth in Ethiopia: Evidence on Cohort 2 PMA-Ethiopia Longitudinal survey (2021-2023).**

| Predictors | Categories | Null model (with no predictors) | Model 1: (Only individual level Predictors) | Model 2 (only community level predictor) | Model 3 (Both individual and community level predictors) |
|---|---|---|---|---|---|
| Age category | 15-19 | Reference | | | |
| | 20-24 | | 1.12(0.53, 2.34) | | 1.09(0.51, 2.33) |
| | 25-29 | | 1.43(0.63, 3.21) | | 1.36(0.59, 3.10) |
| | 30-34 | | 0.75(0.34, 1.63) | | 0.69(0.31, 1.53) |
| | 35-49 | | 0.44(0.16, 1.22) | | 0.40(0.14, 1.13) |
| Gender of newborn | Male | Reference | | | |
| | Female | | 0.55(0.36, 0.83) *** | | 0.53(0.35, 0.82) *** |
| Region | Amhara | Reference | | | |
| | Oromia | | | 0.34(0.13, 0.88) | 0.62(0.30, 1.28) |
| | SNNPR | | | 0.76(0.26, 2.23) | 0.90(0.38, 2.09) |
| | Addis Ababa | | | 0.78(0.28, 2.21) | 2.73(0.90, 8.22) |
| Household Food Insecurity Access category | Food secure | Reference | | | |
| | Mildly Food Insecure Access | | 0.89(0.43, 1.85) | | 0.86(0.42, 1.80) |
| | Moderately Food Insecure Access | | 1.10(0.59, 2.08) | | 1.05(0.56, 2.00) |
| | Severely Food Insecure Access | | 1.59(0.80, 3.13) | | 1.54(0.78, 3.05) |
| Wealth quintile | Poorest | | | | |
| | Poor | | 1.43(0.55, 3.72) | | 1.37(0.52, 3.58) |
| | Medium | | 0.74(0.29, 1.88) | | 0.72(0.28, 1.87) |
| | Rich | | 1.13(0.49, 2.61) | | 1.14(0.48, 2.68) |
| | Richest | | 0.66(0.27, 1.58) | | 0.59(0.24, 1.44) |
| Given tetanus injection during this pregnancy | No | Reference | | | |
| | Yes | | 0.80(0.48, 1.31) | | 0.79 |
| Weight taken as part of ANC | No | Reference | | | |
| | Yes | | 0.99 (0.48, 2.04) | | 0.91(0.44, 1.92) |
| Number of children in this pregnancy | Single | Reference | | | |
| | Twin | | 2.79(0.70, 11.07) | | 2.94(0.72, 12.03) |
| Pregnancy reaction | Very happy | Reference | | | |
| | Sort of happy | | 0.60(0.33, 1.11) | | 0.60(0.32, 1.13) |
| | Mixed happy & unhappy | | 1.14(0.45, 2.87) | | 1.17(0.45, 3.00) |
| | Sort of unhappy | | 0.66(0.23, 1.91) | | 0.67(0.23, 1.97) |
| | Very unhappy | | 1.16(0.30, 4.55) | | 1.26(0.32, 5.03) |
| Partner or husband pregnancy reaction | Very happy | Reference | | | |
| | Sort of happy | | 1.40 (0.78, 2.48) | | 1.38(0.77, 2.48) |
| | Mixed happy & unhappy | | 1.34(0.49, 3.64) | | 1.38(0.50, 3.83) |
| | Sort of unhappy | | 2.43(0.56, 10.59) | | 2.42(0.56, 10.55) |
| | Very unhappy | | 0.67(0.17, 2.59) | | 0.64(0.16, 2.53) |
| Birth preparedness & complication readiness | No | Reference | | | |
| | Yes | | 0.53(0.22, 1.27) | | 0.49(0.20, 1.20) |
| Delivered by cesarian section | No | Reference | | | |
| | Yes | | 1.54(0.79, 3.02) | | 1.45(0.73, 2.88) |

*(Continued)*

**Table 9.** (Continued)

| Predictors | Categories | Null model (with no predictors) | Model 1: (Only individual level Predictors) | Model 2 (only community level predictor) | Model 3 (Both individual and community level predictors) |
|---|---|---|---|---|---|
| Complication during pregnancy | No complication occurred | Reference | | | |
| | occurred complication | | 1.86(1.11, 3.10) * | | 1.89(1.11, 3.20) * |
| Any Delivery complication | No | Reference | | | |
| | Yes | | 1.28(0.82, 1.99) | | 1.30 (0.82, 2.04) |
| Post delivery complication | Not any delivery complication | Reference | | | |
| | At least have one post-delivery complication | | 2.32(1.28, 4.21) * | | 2.33(1.27, 4.27) * |
| Household has an insecticide treated bed net | No | Reference | | | |
| | Yes | | 1.72(1.07, 2.77) * | | 1.81(1.10, 2.96) * |
| Cooking-fuel type used by household | Clean fuel | Reference | | | |
| | Moderately polluting fuel | | 0.94(0.43, 2.04) | | 1.40(0.60, 3.27) |
| | Highly polluting fuel | | 1.66(0.43, 6.41) | | 2.35(0.57, 9.73) |
| Sources of drinking | Improved | Reference | | | |
| | Unimproved | | 1.19(0.57, 2.47) | | 1.18(0.56, 2.48) |
| Fixed effect intercept | | 0.23(0.11, 0.46) *** | 0.30(0.07, 1.27) | 0.37(0.16, 0.83) * | 0.28(0.06, 1.37) |

Note that: *=P value<0.05, **= P value<0.01, ***= indicates P value <0.001.

also indicates that male infants have increased odds of neonatal death and illness (Tura et al., 2014) [46]. The Ethiopian Demographic and Health Survey (EDHS) supports this trend, with slightly higher reported prevalence of common childhood illnesses such as acute respiratory infections, fever, and diarrhea among male children under five, although not always statistically significant. Overall, the observed gender difference in illness prevalence in this study likely reflects inherent biological vulnerabilities in male infants rather than differences in care-seeking or health service access. This underscores the need for heightened clinical vigilance and targeted interventions to reduce preventable illness in this higher-risk group. On the contrary, another study conducted in Burundi revealed that the proportion of under-five hospitalizations that were neonatal was higher among girls than they were among boys: early neonatal infection, prematurity, fetal acute and lung disease [47].

This study found that infants born to mothers who experienced pregnancy-related complications had 89% higher odds of illness (AOR = 1.89; 95% CI: 1.11–3.20), highlighting the substantial influence of maternal health during pregnancy on early neonatal outcomes. Pregnancy-related complications such as hypertensive disorders, antepartum hemorrhage, infections, and gestational diabetes can adversely affect fetal development and compromise neonatal immune function. These conditions are often associated with preterm birth, intrauterine growth restriction, and low birth weight, all of which are well-established risk factors for early-life illness and mortality [48,49]. Maternal complications can also impair placental function, leading to reduced nutrient and oxygen transfer to the fetus, thereby affecting the infant's ability to mount an adequate immune response postnatally [50]. Furthermore, the intergenerational link between maternal and infant health has been emphasized in global health frameworks, such as the WHO's continuum of care model, which advocates for integrated maternal, newborn, and child health (MNCH) interventions to improve survival and well-being across the life course [51].

Similarly, mothers who experienced postpartum complications had more than twice the odds of having an ill infant compared to those without such complications (AOR = 2.33; 95% CI: 1.27–4.27), highlighting the critical role of postpartum care in safeguarding early neonatal health. This finding underscores the critical importance of timely and comprehensive

postpartum care in promoting early neonatal health. The possible justification for this finding might be due to Postpartum complications such as postpartum infections (e.g., endometritis), hemorrhage, severe anemia, and sepsis, can impair a mother's physical ability to provide adequate newborn care, including breastfeeding, hygiene, and timely health-seeking. Disrupted maternal care routines in the immediate postpartum period increase the infant's vulnerability to infectious diseases [52]. Additionally, postpartum illness is a known barrier to exclusive breastfeeding, which is crucial for enhancing infant immunity and protecting against early-life infections [53]. A plausible explanation for the increased risk of infant illness among mothers with postpartum complications is the limited utilization of early postnatal care in settings like Ethiopia. With only 17% of mothers receiving a check-up within the first 48 hours after delivery (EDHS 2016), contributing to missed opportunities for managing complications that affect both mother and child [54].

An unexpected finding was the significantly higher odds of illness among infants residing in households that owned insecticide-treated bed nets (AOR = 1.81; 95% CI: 1.10–2.96). While this appears paradoxical, it likely reflects several confounding factors. Households owning ITNs often reside in malaria-endemic or high-risk environments that are also burdened by poor sanitation, limited healthcare access, and high child morbidity. Moreover, net ownership does not ensure proper or consistent use, infants may be excluded from coverage due to sleeping arrangements, cultural practices, or worn-out nets. Pulford et al. (2011) highlighted substantial gaps between ITN ownership and actual use, especially for infants [55]. Additionally, households with ITNs may be more health-aware and thus more likely to recognize and report illness, introducing potential reporting bias [54]. These findings suggest that ITN ownership may serve as a proxy for environmental risk and health awareness rather than a direct cause of increased infant illness, warranting further investigation into actual usage patterns and contextual exposures.

### Strength and limitation of study

This study's strength lies in its use of weighted, nationally representative data from 1,960 infants aged 5–9 weeks, which enhances the generalizability of the findings to the broader Ethiopian population. Moreover, the analysis appropriately accounted for the complex, hierarchical structure of the data, recognizing that multiple infants may be clustered within the same mother, and mothers themselves are nested within enumeration areas, by applying advanced statistical techniques. This multilevel approach improves the precision and validity of the estimated associations by adjusting for intra-cluster correlation and reducing potential bias. However, reliance on maternal recall to report the presence or absence of infant illness may introduce recall or reporting bias. Additionally, survivor bias cannot be ruled out, as the study includes only infants who survived to at least five weeks of age, potentially underestimating the true burden of early-life illness.

### Conclusion

This study revealed that over one-third of infants aged 5–9 weeks experienced at least one illness episode since birth, underscoring a substantial burden of early-life morbidity. Female infants had significantly lower odds of illness, suggesting a potential biological advantage in early immunity. In contrast, experiencing complications during pregnancy and the postpartum period was associated with higher odds of infant illness since birth, emphasizing the crucial need for integrated maternal care throughout pregnancy, delivery, and the postnatal period. The unexpected association between insecticide-treated bed net ownership and higher infant illness may point to contextual confounders such as greater disease exposure or differential reporting patterns. These findings call for targeted maternal health interventions and further investigation into household-level environmental and behavioral factors influencing neonatal health.

Based on the study findings, we recommend strengthening maternal health services across the continuum of care by improving the early identification and management of pregnancy-related complications, and providing comprehensive postpartum follow-up. Given the increased vulnerability of male infants to early-life illness, integrating neonatal risk screening into routine care is essential. The unexpected association between insecticide-treated bed net ownership and infant illness highlights the need for further research into environmental and behavioral factors influencing newborn

health. Additionally, enhancing community education on maternal and newborn care, proper ITN use, hygiene, and early illness recognition can empower caregivers and improve outcomes. These insights should guide policymakers to design context-specific, gender-sensitive interventions that address both clinical and environmental determinants of neonatal morbidity.

## Acknowledgments

The authors gratefully acknowledge the Performance Monitoring for Action (PMA) Ethiopia project for granting access to the data utilized in this study.

## Author contributions

**Conceptualization:** Amare Mebrat Delie, Mickiale Hailu, Molla Getie Mehari, Gizachew Kassahun Bizuneh, Tesfaye Shumet Mekonnen.

**Data curation:** Amare Mebrat Delie, Tesfaye Shumet Mekonnen.

**Formal analysis:** Amare Mebrat Delie, Mickiale Hailu, Molla Getie Mehari, Tesfaye Shumet Mekonnen.

**Investigation:** Amare Mebrat Delie.

**Methodology:** Amare Mebrat Delie, Mickiale Hailu, Molla Getie Mehari, Gizachew Kassahun Bizuneh, Tesfaye Shumet Mekonnen.

**Software:** Amare Mebrat Delie.

**Supervision:** Amare Mebrat Delie, Tesfaye Shumet Mekonnen.

**Validation:** Amare Mebrat Delie, Mickiale Hailu, Molla Getie Mehari, Tesfaye Shumet Mekonnen.

**Visualization:** Amare Mebrat Delie.

**Writing – original draft:** Amare Mebrat Delie, Mickiale Hailu, Molla Getie Mehari, Gizachew Kassahun Bizuneh, Tesfaye Shumet Mekonnen.

**Writing – review & editing:** Amare Mebrat Delie, Mickiale Hailu, Molla Getie Mehari, Berihun Agegn Mengistie, Tesfaye Shumet Mekonnen.

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
