## [Decision Letter · Decision Letter 0]

14 Nov 2025

Dear Dr. Delie,

Thank you for submitting your manuscript to PLOS ONE. After careful consideration, we feel that it has merit but does not fully meet PLOS ONE’s publication criteria as it currently stands. Therefore, we invite you to submit a revised version of the manuscript that addresses the points raised during the review process.

Please note that we have only been able to secure a single reviewer to assess your manuscript. We are issuing a decision on your manuscript at this point to prevent further delays in the evaluation of your manuscript. Please be aware that the editor who handles your revised manuscript might find it necessary to invite additional reviewers to assess this work once the revised manuscript is submitted. However, we will aim to proceed on the basis of this single review if possible.

We look forward to receiving your revised manuscript.

Kind regards,

Jianhong Zhou

Staff Editor

PLOS ONE

Journal Requirements:

Reviewers' comments:

Reviewer's Responses to Questions

**Comments to the Author**

1. Is the manuscript technically sound, and do the data support the conclusions?

Reviewer #1: Yes

2. Has the statistical analysis been performed appropriately and rigorously?

Reviewer #1: Yes

3. Have the authors made all data underlying the findings in their manuscript fully available?

Reviewer #1: Yes

4. Is the manuscript presented in an intelligible fashion and written in standard English?

Reviewer #1: Yes

Reviewer #1: Thank you for your good work. Now I have attached the comment below.

In Abstract: The results section mentioned the uniformity, some of the sentences used percentages and some of areas use times; is better to maintain the uniformity.

/line 54-56 include the citation.

The research gap or statement of the problem was not identified properly.

Section dependent variable line 122 needs to be paraphrased and clearly mention how to make the category the variable. Again, the author used multiple diseases; if the infant suffered more than one, then how to categorize? As per my suggestion, make the dependent variable a categorical format, with lines for diseases, single, double, and multi-disease.

Need to remove the repetition; there is no difference in the dependent section and the operational definition. Need to be merged.

Riders want to know how to make the wealth quintile (poor….richest). Which variables are the authors taken? Again why author take the same variable in the community poverty status?

In Table 9 few variables are significant, although the author used a 0.25 level of significance instead of 0.05.

The table sentence structure are not uniform.

The author needs to be present sampling design, how to collect the sample, because the author used different region in this study

The author mentioned in the results section that intimate partner violence had higher rates of child, but not mention in the discussion section what are reasons for that.

The author used so many tables, it is better to use the important table in body of the manuscript.

**Do you want your identity to be public for this peer review?** For information about this choice, including consent withdrawal, please see our Privacy Policy

Reviewer #1: **Yes:** Dr. UJJWAL DAS

---

## [Author Response · Author response to Decision Letter 1]

6 Jan 2026

We really appreciate the reviewer comments and constructive feedbacks which help us to improve the quality of our manuscript.

---

## [Decision Letter · Decision Letter 1]

28 Jan 2026

Multilevel Determinants of Illness Since Birth Among Infants Aged 5-9 weeks in Ethiopia: Evidence from the 6-Week Follow-Up of the PMA-Ethiopia Longitudinal Survey (2021–2023).

PONE-D-25-51736R1

Dear Dr. Delie,

We’re pleased to inform you that your manuscript has been judged scientifically suitable for publication and will be formally accepted for publication once it meets all outstanding technical requirements.

Kind regards,

Orvalho Augusto, MD, MPH, PhD

Academic Editor

PLOS One

Additional Editor Comments (optional):

This report has new important data. The authors here use an update survey to report the prevalence of any illness during the second month of life in Ethiopia. For health services provision planning such information is relevant. Differently from many other surveys here the authors leverage a unique longitudinal survey to obtain recent illness events (as they collect information at the age of 2 months). However, this report does not cover what were those illnesses and how frequent they were. Indeed such reporting would take more space.

Few issues to resolve:

1. Why is the whole manuscript in a landscape?

2. There are 2 “figures 1”. Please correct this.

3. Figure 1 - The one showing how the flowchart of the sample - misses the number of neonatal deaths. Is it true that no live birth died before the 5th week of life?

4. Figure 1 - the pie chart - should be removed. It is a misuse of space. This proportion is well covered in the table.

5. There is a “data management and analysis” subsection, then an “Ethical considerations and approval”, and then a subsection specifying the multilevel model. Please, put the specification of the model within the “data management and analysis”.

6. Be careful with the model specification.

• Use an equation editor to avoid terrible symbols. For example: line 214 includes prob(1/ X…). The “/” should be replaced with “|”.

• Title on line 219, I would suggest changing from “random-effect parameters” to something that expresses “interpretation” or “heterogeneity”. We need something more appellative.

• Please avoid using the 2nd person. Eg: line 241 “it tells you how much…” or line 272 “when you add predictors to your model”.

7. Table 3, 4 and 5 - we do not need the “no illness” columns. Ok?

- And why table 3 has a “serial number” column?

Reviewers' comments:

Reviewer's Responses to Questions

**Comments to the Author**

Reviewer #1: All comments have been addressed

2. Is the manuscript technically sound, and do the data support the conclusions?

Reviewer #1: Yes

3. Has the statistical analysis been performed appropriately and rigorously?

Reviewer #1: Yes

4. Have the authors made all data underlying the findings in their manuscript fully available?

Reviewer #1: Yes

5. Is the manuscript presented in an intelligible fashion and written in standard English?

Reviewer #1: Yes

6. Review Comments to the Author

Reviewer #1: The revised manuscript is accepted for production from my side. Just need a minor revision. Athor shold provide the better quality of imgage, it is not visible clearly. And the second methodological need to be shortened, only improtant equation shold be presented, and the remaining must be shifted to suplemantaly file.

7. PLOS authors have the option to publish the peer review history of their article (what does this mean? ). If published, this will include your full peer review and any attached files.

**Do you want your identity to be public for this peer review?** For information about this choice, including consent withdrawal, please see our Privacy Policy

Reviewer #1: No

---

## [Editor Report · Acceptance letter]

PONE-D-25-51736R1

PLOS One

Dear Dr. Delie,

I'm pleased to inform you that your manuscript has been deemed suitable for publication in PLOS One. Congratulations! Your manuscript is now being handed over to our production team.

Kind regards,

on behalf of

Dr. Orvalho Augusto

Academic Editor

PLOS One